# How Instagram Influencers Affect the Value Perception of Thai Millennial Followers and Purchasing Intention of Luxury Fashion for Sustainable Marketing

Akawut Jansom and Siwarit Pongsakornrungsilp *

School of Management, Walailak University, Nakhon Si Thammarat 80160, Thailand; akawut.ja@wu.ac.th
* Correspondence: psiwarit@wu.ac.th; Tel.: +66-75-677-220

**Abstract:** Social media influencers play a significant role in marketing by introducing products to their followers. We investigate how Instagram influencers impact consumer parasocial interaction (PSI) in the relationship between value perception and purchase intention. Whereas customers influence the attractiveness (social and physical) PSI of social media influencers, studies of the effects of luxury purchasing PSI in Thailand are limited. We examine the relationship between PSI and followers of luxury fashion's value (social, personal, and conspicuous) on social media. We use structural equation modeling to evaluate hypotheses by conducting an online survey with 400 Thai millennial respondents who had experience following influencers on Instagram. The findings indicate that Thai millennials accept Instagram influencers' PSIs in terms of value perception and motivation to purchase luxury fashion. The attractiveness of influencers initiates the formation of PSI; followers receive value perception and react to purchasing intention from influencers. The concepts were investigated to prove that influencers' power can encourage followers to mitigate negative consequences by delivering value perceptions on PSI. These findings provided managerial implications for comprehending consumers in the field of digitalization.

**Keywords:** social media; influencer marketing; parasocial interaction (PSI); value perception; millennial consumer; purchasing intention; luxury fashion

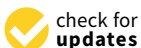



## 1. Introduction

Currently, influencers are being employed by managers to create emotional relationships with consumers through digital marketing [1]. Consumers are living in a consumer culture where socio-cultural factors play an important role in their consumption [2]. Since the revolution of information technology from marketing 1.0 to 5.0, consumers have been able to collaborate and form secondary communities [3] by sharing and interacting through social media and other online platforms [4]. Many brands, celebrities, professional sports players, and so on, have employed social media to build strong relationships with their customers or fans [5,6]. They influence the consumption patterns of consumers in different product categories, including services [7]. This phenomenon can be explained using the concept of parasocial interaction (PSI), which is the interaction of media personalities and media users [5,8]. Online social media, e.g., Twitter, Facebook, and Instagram, play an important role in driving the PSI process between media personalities and consumers [6], which provides an opportunity for marketing scholars to understand how social media can motivate consumers' PSIs.

Recently, PSI has been employed to understand the relationship of PSI and consumption, e.g., understanding sports fan motivation within the interaction between sport and media [5] communication between celebrities and their fans [6], and social media interactions in the vaping community [9]. With strong PSI, consumers can freely access their admired celebrities or influencers and tend to trust the messages from influencers [6]. Additionally, high-PSI consumers tend to follow or share influencers' messages or content

with others [10], which can develop social and physical attractiveness [11,12]. However, other aspects of social media, e.g., violence, bullying, biased opinions toward a particular brand, and deception, might have negative impacts on society [13,14]. Therefore, social media influencers can mitigate the negative impacts of social media by improving PSIs through being positive media personalities.

To mitigate the negative impacts of social media with the concept of PSI, we aimed to understand how PSI can have a positive relationship with value perception and purchase intention. We aimed to identify the factors affecting Instagram influencers that motivate consumers' PSIs and to examine the relationship of PSI with value perceptions of luxury fashion on consumers' purchasing intention [15]. Social and physical attractiveness were also included in this study to explore the relationship of influencers with PSI. Luxury brands were employed as the context of the study because they can express consumer culture consumption and contribute to the generalization of using the concept of PSI development to mitigate the negative aspects of social media influencers. A survey-based quantitative method was employed to collect data from 400 Thai respondents who were Instagram users. SPSS and AMOS were used to analyze the data. Confirmatory factor analysis (CFA) was employed to construct the model and test hypotheses with SEM. To determine the role of PSI in mitigating the negative impacts of social media influencers, we aimed to answer the following research questions:

1. How can an influencer affect followers' value perception and purchase intention to mitigate the negative impacts via Instagram? and
2. How can influencers on Instagram motivate Thai millennials' purchasing intention of a luxury product?

The concepts of social and physical attractiveness, PSI, and value perception were employed to understand their relationship with purchase intention. The findings of this paper contribute to reducing the negative impacts of social media using the relationship between social and physical attractiveness and PSI.

The remainder of this paper is structured as follows: Section 2 provides a literature review, describing influencer marketing before outlining the context of brand luxury in fashion consumption. Section 3 describes the methodology and data collection. Section 4 provides the findings of this study from the questionnaire survey and the hypothesis testing. Finally, Section 5 outlines our recommendations and areas for further research.

## 2. Literature Review

### 2.1. Influencer Marketing in Luxury Fashion Consumption

Opinion leaders acting as social media influencers communicate with a sizeable social network of people who follow them [16]. As the influencer has the ability to sustain consistent sharing habits with followers, such as sharing their lifestyle, tastes, and hobbies [17], they can shape and spread trends. An influencer differs from an entertainer, who maintains an image of themselves. Ryu and Han (2021) [18] stated that as normal people who have no public personas but have strong media personalities, influencers not only persuade their followers but also the public, which differentiates them from entertainers. In contrast with traditional media, social networks provide new accessible functions that allow influencers to interact with large numbers of consumers through communication and relationship building, which affect potential customers' attitudes and behaviors; Choi and Jung (2017) [19] found that 49% of consumers relied on influencer recommendations. Among Korean consumers, the influencers' trustworthiness and expertise positively affect consumers' behavior [20], and influencers' product descriptions and analyses have a strong impact on consumers' product evaluations [21]. Additionally, influencers need to take responsibility for their actions to manage their reputations as being role models for customers [7].

Numerous businesses have implemented influencer marketing strategies to strengthen their ties with consumers by encouraging customer interaction, which involves utilizing social media influencers to convince and stimulate potential buyers to engage with the

brand. One issue here is how influencer marketing can ensure sustainability while also increasing the efficiency and effectiveness of social media advertising campaigns. Recent research [7] showed that an influencer and marketing relationship can serve corporate sustainability. The authors discovered that taking into consideration potential influencers' emotional projection abilities when determining brand sponsorship and target addressee is important in social media advertising on the concept of corporate sustainability. This led to research question 1: How can an Instagram influencer persuade followers' value perception and purchase intention to mitigate the negative impacts via Instagram?

Luxury brands are a valuable market for millennial consumers, whose purchasing intentions are often influenced by social media, such as Instagram [22]. With the increasing demand for luxury items, customers are becoming more fashion sensitive and buying behaviors are especially affected by fashion trends [23]. Luxury brands are followed by fashion-conscious, younger targets, who are normally affected by fashion influencers or fashion leaders on social media [24]. Fashion content and social media can affect followers' opinions and purchase behavior, as "high-quality, expensive, and non-essential goods and services, luxury brands appear to be rare, unique, historic, and worthy, and provide consumers with high levels of symbolic and emotional/hedonic value" [25].

As mentioned by Vargo and Lusch (2016) [26], customers play an important and active role in the value co-creation process as a value co-creator. In terms of value and consumption, Vigneron and Johnson (2004) [27] described luxury brands as prestige brands with a high degree of perceived non-personal characteristics (e.g., conspicuousness, exclusivity, and quality) and perceived personal characteristics (e.g., hedonism and extended self). To qualify as luxurious, brands must exhibit luxury characteristics (e.g., prestige, superior quality, high price, exclusivity, and uniqueness) and deliver psychological and emotional value, especially conspicuous and hedonic value. Many scholars [28,29] recognized that fashion products especially have high esthetic and hedonic value. Most consumers feel that fashion products have high social and symbolic value. Accordingly, fashion products are able to build psychological and emotional value similar to luxury products when they are expensive, high quality, and rare.

In consumption culture, luxury products offer exclusivity and prestige to high-society consumers. Luxury brands, e.g., Chanel, Dior, Burberry, Gucci, and Prada, have implemented social media marketing to engage consumers [30]. For instance, Burberry reported a profit of 40% using social media marketing. Chu et al. (2013) [31] found that developing a marketing strategy for luxury brands for social media can increase consumers' favorable perceptions, desire for luxury, and purchase intentions. In social media, an influencer is described as someone who posts content for remuneration. [32]. Lou and Yuan (2019) [33] stated that a social media influencer is a person who is first and foremost a content creator: someone who possesses expertise in a particular field and has built a sizable following of captive followers who add marketing value to a brand through consistently producing valuable content on social media. Hence, influencer marketing is one of the strategies that can motivate customers' purchase intentions; therefore, many luxury brands unite on social media platforms. However, there is the widespread belief that social media is a threat to luxury brands' reputations and status; it aids in the development of broad appeal but can detract from the selling point of premium goods [34]. Because luxury brands are increasingly embracing social media and influencers to communicate with their customers [35], it is vital to understand how these luxury brands enhance consumer engagement on social media.

### 2.2. Parasocial Interaction (PSI) and Mitigating Effect of Online Violence

The relationship between media personalities and media users is called PSI [5,36]. Horton and Wohl's (1956) [36] definition of PSI is that media users establish a relationship with media personalities over time through group communication. In contrast with social interaction, PSI is a one-sided and mediated interaction where online users frequently actively rather than passively engage in their relationships with media personas [37]. Rela-

tionships between media personalities and media users are often experienced as "seeking guidance from media personae, seeing media personalities as friends, imagining being part of a favorite program's social world, and desiring to meet media performers" [12].

PSI is considered a friendship through which media users seek representation from media personalities as if they were friends, despite media users indirectly having a connection with media personalities [12]. Moreover, individuals tend to imagine the persona as a real friend [38,39]. PSI is comparable to interpersonal relationships and can act as a functional alternative for them; the voluntary nature, providing companionship, and social attractiveness are factors in establishing these relationships [40]. Online social media created a new area of academic research on PSI by providing the opportunity for the investigation of two-path communication and balanced relationships [6]. Many studies of influencers have indicated that interactions mostly go one way. Although online social media, such as Instagram, freely allow their fans access to the personal lives of the media personalities, it is a reactive experience as a one-sided relationship [5].

Melović et al. (2020) [41] analyzed attitudes toward online violence and identified Instagram as the main platform for violence on online media; in particular, social marketing may play a key role in the prevention of online violence. Stever and Lawson (2013) [6] found that influencers market by acting as role models for followers through creating positive social change. Influencers are opinion leaders who can impact social media forums through their interactions with their followers. Influencers can cause their fans to be infatuated with them through interesting messages, photos, captions, and sharing content because customers have unrestricted access to their favorite celebrities or influencers and they are more likely to trust the messages delivered by influencers. The positive PSI of an influencer and follower might reduce the negative social media impacts, such as violence, bullying, biased opinions toward a particular brand, and deception [13,14]. Powerful influencers on social media are the key to urging followers to engage socially or in social activities [42] through their posts. Therefore, we employed PSI to understand how the relationship between an influencer and a follower can mitigate the negative impacts of social media on luxury brand consumption.

Numerous studies have considered luxury brands and online social media, but research in Thailand is limited [43,44]; therefore, we aim to fill this gap by studying the relationship between PSI and purchase intention with Thai consumers, who are a potential market for luxury brand products. Lee and Watkins (2016) [45] explored how YouTube video blogs motivate consumer perceptions of luxury brands, brand perceptions, and purchase intentions using experimental groups. To achieve marketing goals, influencers should possess powerful convincing skills to persuade followers [46]. However, studies are lacking on the relationship between influencers and purchase intention in luxury fashion. This led to the second research question: How can influencers on Instagram motivate Thai millennials' purchase intentions of luxury products?

### 2.2.1. Attractiveness of Instagram Influencers

Users perceive information differently depending on their characteristics. The online influencers who persuade followers via posts, i.e., images, wording, and short videos, may play a role as a source of attraction [47], meaning that the image of a person (e.g., an actor, presenter, or celebrity) may act as a draw for the audience in order for them to comprehend the message being communicated. Media personalities attract social and physical attention, which can be used to drive PSI [38]. Additionally, attraction to a media personality can enhance the number of rewarding interactions or reiterate surveying. When online users perceive a media personality who is similar to themselves and others in their interpersonal network, the process of PSI emerges, through which consumers tend to follow or share the content of that media personality [11]. Social attractiveness can be defined as the possibility of befriending or selecting a media person as a social or work partner [15]. PSI has been demonstrated to be a predictor of social attractiveness for both traditional media (television and newspapers) and new media (social media) [48,49]. A

recent study of French users reported a negative relationship of physical attractiveness in supporting users' PSIs and purchase intention within beauty and fashion contexts, whereas social attractiveness positively supported the relationship between social media influencers and users [15]. However, the theoretical grounds of previous research examining social attractiveness is the main factor that influences PSI (e.g., Rubin and McHugh, 1987 [48]; Lee and Watkins, 2016 [45]; and Kurtin et al. (2018) [49]). Therefore:

**Hypothesis 1 (H1).** *The social attractiveness of an Instagram influencer increases PSI.*

Physical attractiveness relates to a persona's physical appearance, e.g., beauty vloggers with attractive facial features or an attractive physical appearance [47]. Online influencers who demonstrate the attractive appearance of products play an important role in persuading more followers to watch such content. As a result, followers develop a sense of affinity toward them and consider them as friends [45]. In other words, followers will follow all the content the influencer shares, e.g., images, short videos, and captions. Based on the findings of previous research examining physical attractiveness is the factor influence to PSI (e.g., Lee and Watkins, 2016 [45], Purnamaningsih and Rizkalla, 2020 [47], Sokolova and Kefi, 2020 [15], and so on), we proposed the following hypotheses for identifying the antecedents of PSI:

**Hypothesis 2 (H2).** *The physical attractiveness of an Instagram influencer increases PSI.*

2.2.2. Effect of PSI on Consumer Value Perception of Luxury Brands

As mentioned by Vargo and Lusch (2016) [26], customer value is a benefit that customers gain from using, possessing, or consuming products and services that are different in time, situation, person, and place. Relationships between PSI, a media personality, and users on Instagram emerge when users engage with the media personality (Rubin et al., 1985) [12]. Luxury consumers can perceive conspicuous value, social value, and quality value when buying luxury products [50]. When a consumer considers purchasing luxury products, they automatically send a status signal to others in society [51,52]. Therefore, users who seek and perceive luxury products from a media personality as an influencer will express value perception to buy products and signal their social status.

As discussed above, the concepts of luxury align with customer value, which varies from person to person depending on their social status, ethnic group, background, culture, and the personal consumption experiences of each individual [44,53,54]. Due to the multiple dimensions of value [3], we employed value perception, social value, personal value, and conspicuous value to study PSI's effect on consumer perceptions of luxury brands. According to Smith and Colgate (2007) [55], luxury value perception is an outstandingly strong predictor of luxury purchases. Luxury value perception is a belief used to guide the selection or evaluation of desirable products and to make decisions regarding buying a particular product [56]. A consumer's sense of luxury value can clarify exactly why they purchased a particular luxury item [57].

In the context of luxury brands, through upward perception value, users perceive higher value for luxury brands after viewing short videos and photos on Instagram, which probably enhances the desire to purchase luxury products. Hence, we expected positive effects of PSI on social value and conspicuous value. Social value is undoubtedly a factor when consumers purchase luxury goods in the expectation of enhancing their personal status [58] as a result of the prestige associated with social status [59]. Additionally, status is a process through which individuals who purchase luxury products increase their social image level [60]. Zhan and He (2012) [61] underlined that understanding the problem of luxury consumption is critical in identifying the social norms that indicate who are luxury consumers. Thai society is a collectivist society [62], which offers a perspective on how different cultures regard social norms. The author also conducted research within Thailand regarding Thai consumers' behavior when purchasing luxury products. Social motivation was found to positively affect willingness to purchase luxury brands [63].

Consumers are mostly concerned with aspects such as physical appearance (physical vanity) and notice a positive impact of their vanity on their personality [64]. Consumers purchase a luxury product because it can help them to achieve their life goals and create the consumer's identity [65]. Social visibility is determined prominently by the comfortable feeling when buying a product or service [66]. Previous research consistently reports that positive perceptions of social value increase consumer purchase intentions (e.g., Cheah et al. (2015) [63], Netemeyer et al. (1995) [64], Cesare and Gianluigi, 2011 [65], and so on), therefore, we constructed the following hypotheses:

**Hypothesis 3a (H3a).** *PSI with Instagram influencers increases positive social value perceptions of luxury brands (like Gucci).*

**Hypothesis 4a (H4a).** *Social value perceptions of luxury brands (like Gucci) positively motivate consumers' reaction to purchase intentions.*

Personal value is the concept of creating self-projection through objects that are perceived by the individual as correlating with one's attitudes, feelings, perceptions, and evaluations [44], whereas PSI is perceived and expressed via stimulating product purchases. Personal value is how a person thinks others perceive them, and product possession can significantly contribute to and reflect their identities [44,67–69]. Consumers have the capacity to express themselves through their styles and product purchases. Therefore, based on previous research showing that positive perceptions of personal value increase consumer purchase intentions (e.g., Oe et al. (2018) [44] and O'Cass and Siahtir (2013) [69]), we constructed the following hypotheses:

**Hypothesis 3b (H3b).** *PSI with Instagram influencers increases positive personal value perceptions of luxury brands (like Gucci).*

**Hypothesis 4b (H4b).** *Personal value perceptions of luxury brands (like Gucci) positively motivate consumers' reaction to purchase intentions.*

Conspicuous consumption contributes to the formation of preferences for a wide variety of products. The term conspicuous consumption was first used by Veblen in 1902 in *The Theory of the Leisure Class*. Owning luxury goods can display wealth and social status to others, which are referred to as luxury products [70]. As an example, O'Cass and Siahtir (2013) [71] found that Chinese young adults prefer Western fashion brands because they tend to represent status and wealth more effectively than Asian brands. The theoretical grounds of Veblen (1903) [51] suggest that positive perceptions of conspicuous value increase consumer purchase intentions; therefore, we constructed the following hypotheses:

**Hypothesis 3c (H3c).** *PSI with Instagram influencers increases positive conspicuous value.*

**Hypothesis 4c (H4c).** *Conspicuous value perceptions of luxury brands (like Gucci) positively motivate consumers' reaction to purchase intentions.*

2.2.3. Effect of PSI on Consumer Purchase Intention

Purchase intention is "an individual's conscious plan to make an effort to purchase a brand" [72]. Kim and Ko (2012) [73] stated that brand attitude is one of summative evaluation, whereas purchase intention is a personal tendency correlating to a brand with an intention of implementing buying behavior. Over the past few years, social media platforms have increasingly received attention from marketers seeking to capitalize on opportunities to persuade consumers to purchase. An important component that can influence the likelihood of consumer purchasing intention via social media platforms is

electronic word of mouth [74,75]. Another crucial component is the social media influencer's credibility in an endorsement situation [15].

Horton and Wohl (1956, p. 5) [36] stated that "if individuals were immersed in PSI, they would "contribute to the illusion by believing in it, and by rewarding the persona's 'sincerity' with 'loyalty'" and, through this, their behaviors might be influenced." In the offline context, there are extensive studies of the effect of PSI on consumer behavior. Stephens et al. (1996) [76] determined that spectators are more likely to buy products via a TV shopping program that has an identical host with whom viewers have established a relationship, where they tended to be convinced to purchase unanticipated goods. In contrast, social media platforms make it easier to initiate and enhance users' PSIs. In China, Zheng et al. (2020) [10] found that the perception of PSI in social media is positively associated with the user's social commerce intention. However, media personalities on social media platforms encourage users to seek, follow, share, and comment about content to others who have similar interests, styles, and shopping goals. Individuals' PSI with other users causes them to accept their recommendations as if they were real-world friends and determines their subsequent behavioral intentions. Additionally, the literature regularly reports that positive brand attitudes and perceptions increase consumer purchase intentions (e.g., Bian and Forsythe (2012) [77]; Kim and Ko (2012) [73]; Zhang and Kim (2013) [78]; Lee and Watkins (2016) [45]; and so on). Therefore, we proposed the following hypothesis: See Figure 1.

**Hypothesis 5 (H5).** *PSI with Instagram influencers are a positive reaction associated with the intention of purchasing luxury brands (like Gucci).*

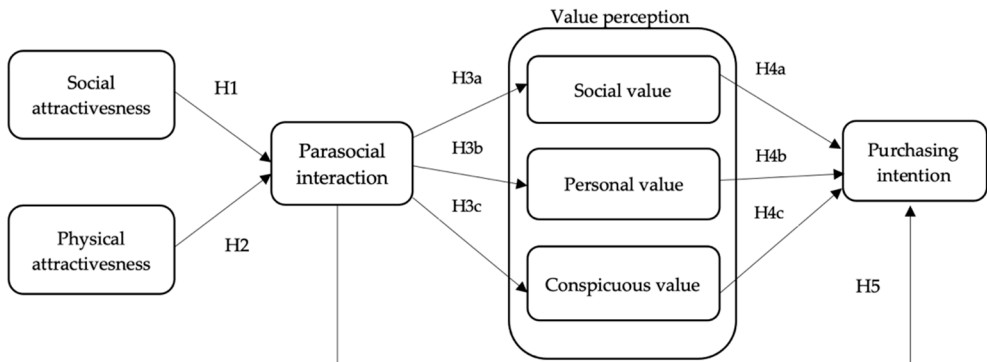

**Figure 1.** Conceptual framework of this study.

## 3. Methodology

### 3.1. Data Collection and Samples

We used a deductive approach related to this research's aim of testing and validating the conceptual model, which involved administering a survey to collect data to investigate our hypotheses that were formulated based on our literature review [79]. Based on the results of the hypotheses testing, conclusions were drawn to confirm the nature and degree of alignment between the theory and the gathered data. The data were collected in March 2021. The study was conducted using a questionnaire-based online survey on Google Forms with closed-ended questions using a seven-point Likert scale (1 strongly disagree to 7 strongly agree), all of which were adapted from previous studies to quantify the responses. The 34 questions were translated into Thai together with back translation from scholars in the marketing field ensuring content validity, which included questions for demographic and hypothesis variables: social attractiveness was measured on a four-scale item validated in previous research on social shopping context [10]. Both physical attractiveness and PSI variables were adapted using a four-item scale validated in previous research into YouTube influencers [45]. Value perceptions including social value variable, personal value variable and conspicuous value variable were developed using a four-item scale validated into

luxury product context from previous research [44], and purchase intention was measured on a four-item scale adapted from [45].

The sample was gathered by sending an online questionnaire to social media platforms, including Facebook group, Instagram, and LinkedIn. The sampling procedure was non-probabilistic (convenience) and included 408 Thai respondents aged between 23 and 39 years who had experiences with following Instagram influencers. After excluding invalid responses, we had a sample of 400 questionnaires for the analysis as Table 1, the majority of the sample (56.25%) was male, the average monthly income was 40,000 Thai baht, and master's degrees were the highest educational level of the participants (55.75%). The respondents spent 3–5 h per day on social media and had between 701 and 800 Instagram followers on average. Of the sample, 86.25% of Thai Millennials consider Instagram (IG) influencers when buying luxury fashion.

**Table 1.** Demographic characteristics.

| | *n* | % | | *n* | % |
|---|---|---|---|---|---|
| Gender | | | Education | | |
| Male | 255 | 56.25 | High School | 5 | 1.25 |
| Female | 105 | 26.25 | Bachelor's degree | 116 | 29 |
| LGBTQ | 60 | 15 | Master's degree | 223 | 55.75 |
| Prefer not to say | 10 | 2.5 | Doctoral degree | 56 | 14 |
| Income per month (Thai baht) | | | Time spent on social media per day | | |
| <20,000 | 21 | 5 | <1 h | 20 | 5 |
| 20,000–30,000 | | | | | |
| 30,000–40,000 | 71 | 18 | 1–2 h | 129 | 32.25 |
| >40,000 | 112 | 28 | 3–5 h | 164 | 41 |
| Followers on Instagram | 196 | 49 | 6–8 h | 64 | 16 |
| <500 | 90 | 22.50 | 9+h | 23 | 5.75 |
| 501–600 | 69 | 17.25 | Does an IG influencer(s) affect your | 345 | 86.25 |
| 601–700 | 56 | 14 | considerations when buying luxury fashion? | 55 | 13.75 |
| 701–800 | 135 | 33.75 | Yes | | |
| >800 | 50 | 12.50 | No | | |

### 3.2. Analysis

As a result, we provide three research measures that were calculated using SPSS 17.0 and AMOS 21.0. The first phase involved describing the variables of the model through which the influencers persuade consumers to purchase luxury products; the second phase involved confirmatory factor analysis (CFA), which was done to verify the adequacy of the items for the factors and the number of dimensions underlying the model construction in this empirical model, where SEM was used to test the relationships between the various constructs.

Sample: *n* = 400 respondents.

### 3.3. Validation of Measures

To ensure the validity and reliability of the questionnaires regarding how influencers persuade consumers to purchase luxury products, the measured variables included social attractiveness, physical attractiveness, parasocial interaction, social value, personal value, conspicuous value, and purchase intention. The results showed that reliability testing ranged from 0.811 to 0.878 and the overall Cronbach's alpha was 0.939, which is greater than the standardized recommendation of 0.70 by Nunnally and Bernstein (1994) [80]; therefore, the questionnaire in this research can be considered valid, as demonstrated in Table 2.

**Table 2.** The reliability of the variables through which influencers persuade consumers to purchase luxury products.

| Variable | Cronbach's Alpha |
|---|---|
| Social attractiveness | 0.814 |
| Physical attractiveness | 0.844 |
| Parasocial interaction | 0.811 |
| Social value | 0.878 |
| Personal value | 0.806 |
| Conspicuous value | 0.856 |
| Purchase intention | 0.877 |
| Overall | 0.939 |

All measured on a 7-point Likert scale, *n* = 400.

## 4. Results and Discussion

### 4.1. Descriptive Statistics

Table 3 provides the descriptive statistics that were used to analyze the variables of the model in order to demonstrate how influencers persuade consumers to purchase luxury products. The results show that all standard deviations were less than 1.5 (30% of the mean); therefore, the data were not widely dispersed from the mean, with a mean range of 3.10–4.69 and a standard deviations range of 1.09–1.64.

**Table 3.** Descriptive statistics of the variables through which influencers persuade consumers to purchase luxury products.

| Variable | Mean | SD |
|---|---|---|
| **Social Attractiveness** | | |
| SA1: Instagram influencers have many things in common with me. | 3.89 | 1.38 |
| SA2: They are similar to me. | 3.85 | 1.24 |
| SA3: They share my worth. | 3.96 | 1.20 |
| SA4: They have thoughts and suggestions that are close to mine. | 3.81 | 1.25 |
| **Physical Attractiveness** | | |
| PA1: Instagram influencers look physically attractive. | 4.69 | 1.64 |
| PA2: Their lifestyles are physically attractive. | 4.27 | 1.32 |
| PA3: I think he/she is quite pretty or handsome. | 4.68 | 1.42 |
| PA4: I think he/she is very sexy looking. | 4.51 | 1.36 |
| **Parasocial Interaction** | | |
| PS1: When you see an influencer use a product in daily life on their Instagram, you look forward to purchasing that functional product. | 3.69 | 1.20 |
| PS2: At first, you did not want to buy this product but when you saw they used it, you really wanted to buy it. | 3.73 | 1.12 |
| PS3: I thought that socializing was not necessarily about using brand-name products, but when the influencer did it, I felt that the brand-name products were also a requirement for me. | 3.64 | 1.11 |
| PS4: You decided to choose Gucci when you saw influencers used Gucci. | 3.64 | 1.16 |
| **Social Value** | | |
| SV1: Influencers can influence your luxury purchase decisions. | 3.63 | 1.14 |
| SV2: For you, social status is really important. | 3.68 | 1.13 |
| SV3: By using luxury brands (Gucci) similar to them, you think it can demonstrate your social status as an influencer. | 3.66 | 1.12 |
| SV4: If you use Gucci, it can increase your social confidence. | 3.73 | 1.14 |
| **Personal Value** | | |
| PV1: Luxury goods improve your self-confidence. | 3.55 | 1.21 |
| PV2: You would like to purchase items different to what others purchase to display your individuality, which can help to improve your image. | 3.55 | 1.16 |
| PV3: You think you should signal your social standing and wealth by purchasing Gucci products. | 3.55 | 1.09 |
| PV4: You think that luxury products can fulfil your intangible needs. | 3.61 | 1.20 |

**Table 3.** *Cont.*

| Variable | Mean | SD |
|---|---|---|
| Conspicuous Value | | |
| CA1: When you purchase luxury brands, you want to display wealth but you do not care about the quality of the product. | 3.35 | 1.11 |
| CA2: Sometimes you buy a product because most people use it, but you do not even think about its suitability. | 3.10 | 1.15 |
| CA3: You believe that purchasing Gucci can differentiate you from non-prestigious groups. | 3.43 | 1.12 |
| CA4: When Gucci launches a new seasonal collection, I usually buy the products even though this will impact my finances in the future, perhaps because the products will allow me to gain respect from others. | 3.35 | 1.09 |
| Purchase Intention | | |
| PI1: Purchasing a brand name increases your societal standing. | 3.58 | 1.10 |
| PI2: If I see the Gucci brand when I go shopping, I will buy it. | 3.60 | 1.19 |
| PI3: On special occasions, I will make an effort to buy Gucci for special people. | 3.55 | 1.12 |
| PI4: In the coming years, I hope to buy luxury fashion products. | 3.62 | 1.20 |

*4.2. Confirmatory Factor Analysis (CFA)*

This section provides the CFA results of the model describing how influencers persuade consumers to purchase luxury products. The objective of CFA is used to determine the suitability of items for factors and the number of dimensions in an empirical model. [81], and to identify dependent variables. CFA is conducted to examine the fit of the data to the empirical study [82]. In this study, we chose a theoretical framework with seven variables: social attractiveness, physical attractiveness, parasocial interaction, social value, personal value, conspicuous value, and purchase intention.

The results of the multi-factor confirmatory analysis in this model show the acceptable threshold levels, which is consistent with Hair et al. (1998) [83], Bollen (1989) [81], and Sorbon (1996) [84]. The result shows that the chi-square was 187.216 (df = 187.00, Sig. = 0.482 > 0.05, CMIN/df. = 1.001 < 2.0). Table 4 provides the CFA results of the model explaining how influencers persuade consumers to purchase luxury products. The results show that the squared multiple correlations ($R^2$) ranged from 38.0% to 90.0%, and the standardized factor loading ranged from 0.61 to 0.95, which is higher than the 0.50 threshold recommended by Barclay et al. (1995) [85]. The average variance extracted (AVE) quantifies the variance collected by the indicators in relation to the measurement error; we found that it ranged from 0.508 to 0.627, which is higher than 0.50. The composite reliabilities (CRs) for all constructs in the model were above the threshold value of 0.70 [86,87]. We found that the composite reliability ranged from 0.788 to 0.870, which is higher than 0.60. All variable measurements were found to be acceptable and strongly suggested that each set of items represents a single underlying construct and provides evidence for discriminate validity or fit confirmation. Overall, the findings indicated an excellent fit to the data using the testing model (See Table A1 on Appendix A).

**Table 4.** Results of path SEM.

| | Variable | Path | Variable | Coefficient | S.E. | t. | Sig. | $R^2$ |
|---|---|---|---|---|---|---|---|---|
| H1 | Parasocial interaction | ← | Social attractiveness | 0.440 | 0.060 | 6.886 | 0.000 * | 28.0% |
| H2 | Parasocial interaction | ← | Physical attractiveness | 0.150 | 0.040 | 2.597 | 0.009 * | 28.0% |
| H3a | Social value | ← | Parasocial interaction | 0.870 | 0.090 | 14.173 | 0.000 * | 75.0% |
| H3b | Personal value | ← | Parasocial interaction | 0.890 | 0.080 | 10.941 | 0.000 * | 79.0% |
| H3c | Conspicuous value | ← | Parasocial interaction | 0.690 | 0.080 | 10.670 | 0.000 * | 48.0% |
| H4a | Purchase intention | ← | Social value | 0.330 | 0.070 | 4.078 | 0.000 * | 70.0% |
| H4b | Purchase intention | ← | Personal value | 0.430 | 0.150 | 3.433 | 0.000 * | 70.0% |
| H4c | Purchase intention | ← | Conspicuous value | 0.130 | 0.040 | 2.965 | 0.003 * | 70.0% |
| H5 | Purchase intention | ← | Parasocial interaction | 0.030 | 0.220 | 0.153 | 0.879 | 70.0% |

* Sig. < 0.05.

### 4.3. Structural Equation Modeling

Structural equation modeling (SEM) has become one of the data analysis techniques used by researchers across various disciplines for maximum likelihood estimation. Therefore, we analyzed the constructed model using SEM analysis to test the relationships between the various constructs. The chi-square value is the traditional measure for evaluating the model's overall fit and assessing the magnitude of the discrepancy between the sample and the fitted covariance matrices [88]. A good model fit provides a significant result at a 0.05 threshold [89]. The result was found to be acceptable and consistent with the concepts of Hair et al. (1998) [83], Bollen (1989) [81], and Sorbon (1996) [84]. The chi-square was 230.543 (df = 218.0, Sig. = 0.267 > 0.05, CMIN/df. = 1.058 < 2.0). The results of the SEM for analyzing how the influencers persuade consumers to purchase luxury products demonstrated a reasonable fit of the index of model based on several fit statistics. See Figure 2.

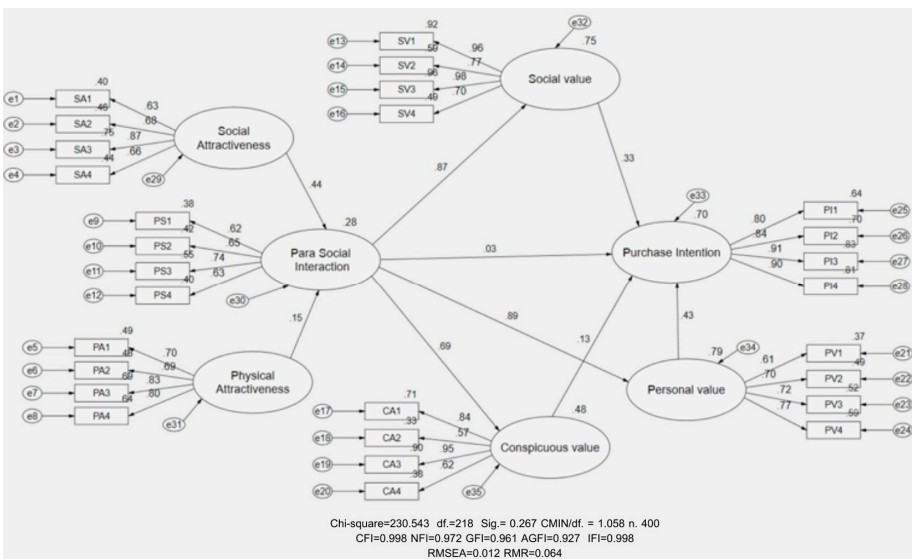

**Figure 2.** Structural equation modeling analysis results.

### 4.4. Hypothesis Testing

Regarding the SEM analysis in Table 4, the data reveal the hypothesis testing analysis results for the fit confirmation of the model of how influencers persuade consumers to purchase luxury products, summarizing the path coefficients and the hypotheses.

H1 showed that the social attractiveness of an Instagram influencer strongly increases PSI, with a regression weight estimate of standardized coefficients of 0.440, standard error of about 0.060, *t*-value 6.886, and Sig. 0.000 < 0.05. Influencers' social attractiveness influenced the change in the increased PSI, with the model explaining the influence on change at a rate of 28% (H1 supported, Sig = 0.000 *). The result agrees with Sokolova and Kefi (2020) [15] who examined beauty and fashion influencers in France and found that social attractiveness influences people to create PSI. The result is also in line with the findings of both Perse and Rubin (1989) [38] and Purnamaningsih and Rizkalla (2020) [47]: followers perceiving media personalities as similar to themselves leads to PSI. Meanwhile, H2 showed that the physical attractiveness of Instagram influencers strongly increases PSI, with a regression weight estimate of standardized coefficients of 0.150, standard error of about 0.040, *t*-value 2.597, and Sig. 0.009 < 0.05; influencers' physical attractiveness influenced the change in the increased PSI, with the model explaining the influence on change at a rate of 28% (H2 supported, Sig = 0.009 *). The result shows that a person's physical appearance plays an important role with followers, as noted by Purnamaningsih and Rizkalla (2020) [47]. Thai followers build a relationship with influencers who have attractive facial features and regard them as online friends [45]. The findings of H1 and H2 are consistent with

existing research on the attractiveness of leaders' social media personalities which contends that both social and physical attractiveness persuade a consumer to have PSI [38,45]. The findings are also in line with Stever and Lawson (2013) [6] and Zheng et al. (2020) [10] who claim that strong PSI can encourage consumer trust through influencer messages. However, Table 4 indicates that social attractiveness is a stronger indicator of PSI than physical attractiveness. The study collected the interaction opinions of millennial consumers who follow social media personalities like Instagram influencers and found these followers are persuaded more through social attractiveness than physical attractiveness, which accords with previous research [15,45]. An Instagram influencer who presents information that followers find similar to themselves can lead to greater PSI than the appearance of an influencer. Thus, both the social and physical attractiveness of an Instagram influencer increases PSI which can motivate followers to engage socially or on social media and be utilized in building long-term relationships with customers as a viable marketing technique [90].

Moreover, the findings showed that PSI with Instagram influencers increases the positive perception of luxury brands' social value through their posts, with a regression weight estimate of standardized coefficients 0.870, standard error of about 0.090, *t*-value 14.173, and Sig. 0.000 < 0.05. PSI had a 75.0% influence on the increase in Gucci's social value (H3a supported, Sig. 0.000 *). The result indicates that Thai consumers who have PSI with influencers expect luxury goods to enhance their personal status and consider purchasing them due to the prestige associated with social status [58,59]. The findings also showed that PSI with Instagram influencers through their posts increases positive value perception of the personal value of luxury brands, with a regression weight estimate of standardized coefficients 0.890, standard error of about 0.08, *t*-value 10.941, and Sig. 0.000 < 0.05. Additionally, PSI influenced Gucci's positive personal value change at 75.0% (H3b supported, Sig. 0.000 *). Thai consumers who are influenced by PSI consider their own purchasing decisions when it comes to luxury goods. This relates to the concept of self-projection through objects that an individual perceives as correlating with his or her attitudes, feelings, perceptions, and opinions [44]. The findings indicate that PSI with Instagram influencers through their posts increases positive value perception of the conspicuous value of luxury brands, with a regression weight estimate of standardized coefficients 0.690, standard error of about 0.08, *t*-value 10.670, and Sig. 0.000 < 0.05. PSI also influenced Gucci's positive conspicuous value change at 75.0% (H3c supported, Sig. 0.000 *). PSI with influencers persuades consumers to consider the conspicuous value of buying luxury products to display wealth and social status to others [70]. Thai followers who have PSI with Instagram influencers perceive value through their posts, and thus influencer messages play an important role in stimulating online society which might lead to mitigating online effects through perceived value perceptions of followers.

Value perceptions of luxury brands positively motivate consumer purchase intentions through social value (H4a), personal value (H4b), and conspicuous value (H4b). Thai consumers' intentions to purchase luxury brands are positively impacted by social value perceptions (H4a supported, Sig. 0.000 *). Social motivation has been found to have a positive effect on one's willingness to purchase luxury brands. In China, consumers are critical in identifying the social norms that indicate who are luxury consumers [61]. The result also showed that personal value perceptions of luxury brands positively motivate consumer purchase intentions (H4b supported, Sig. 0.000 *), even more than social and conspicuous value. A person's self-perception is influenced by how others perceive them, and product ownership significantly contributes to and reflects one's identity [44,68,69]. Relatedly, Thai consumers' purchase intentions of luxury brands react positively based on conspicuous value perceptions (H4c supported, Sig. 0.000 *). The result showed that a factor in their purchase intentions is to display wealth and social status to others [70]. However, previous research by Oa et al. (2018) [44] reported that conspicuous value does not appear to be a significant factor in determining purchase intention for luxury handbags. As this paper investigated the online context, perhaps the value perceptions of

influencers persuade Thai millennials. Consequently, Thai millennial consumers who buy luxury fashion items tend to make purchase decisions influenced by a collectivist culture and prefer visible public luxuries to deliberately express their social status. These are highlights to prove how follower intention to purchase via perceived PSI value perceptions may mitigate online effects or social movements via reaction after perceiving PSI value perceptions to demonstrate the follower's social status on public social media platforms.

Interestingly, PSI with Instagram influencers is not a significant factor associated with purchasing intention of luxury brands (like Gucci) (H5 unsupported). The result contrasts with Kim and Ko (2012) [73] and Zhang and Kim (2013) [78], who found that PSI in social media motivates user buying intention. However, the study showed that Thai respondents determine value perception prior to forming purchasing intention for the luxury brand. Lang and Armstrong (2018) [23] support the finding that when consumers intend to purchase an expensive item, they consider the value perceived by society. Therefore, in a luxury brand context, PSI with Instagram influencers does not directly motivate Thai followers' purchase intentions without the intermediary of value perception.

The above discussion emphasizes the importance of studying luxury products. When purchasing luxury products, Thai consumers listened to an influencer. The study achieved its main objective by determining that engaging with an influencer through PSI motivates value perceptions that impact purchase intentions of luxury products. Through social and physical attractiveness, an Instagram influencer encourages follower PSI and provides perceptions of social, personal, and conspicuous value that affect follower purchase intention. These are the key points in how PSI can socially motivate social media users, and these results potentially mitigate the negative impression of social networks' impact through studying the concept of luxury consumption.

## 5. Conclusions and Implications

By studying value perception's relationship to purchase intention, our conclusion sheds light on how PSI with influencers can be employed to mitigate the negative impacts of social networks. We extended Oa et al. (2018) [44] by investigating the influence of social media leaders on luxury fashion perceptions and purchase intentions. The way Instagram influencers motivate consumer PSI was identified and employed to examine the relationship between the PSI with Instagram influencers, luxury fashion value perceptions, and consumer purchasing intention to support sustainable marketing. The results clarify how PSI with influencers motivates followers' value perceptions (social, personal, and conspicuous value) and impacts their purchase intentions. This concept can support mitigation of the negative impact of social network sites on influencers' posts. Therefore, we can conclude with three points relating to the study's objective and managerial implications.

This study showed that followers positively perceive both social and physical attractiveness, but Thai millennials recognize the social attractiveness over the physical attractiveness of media personalities. Thai millennials primarily follow media personalities whom they see as similar to themselves rather than choosing them based on appearance. The study indicates that Thai millennials who follow Instagram influencers might base their value perceptions, online activity, and trust on messages from these media personalities, leading to immersion in PSI [36].

Furthermore, PSI with Instagram influencers positively motivates followers' value perceptions of luxury brands. When Thai millennials have PSI with media personalities, they perceive value when considering purchases, including social, personal, and conspicuous value. The study showed that Instagram influencers create personal value perception, which reaches the highest level when followers encounter a trend via influencers and consider purchasing an expensive item on their own. While social and conspicuous value perceptions are created in PSI with Instagram influencers, Thai millennial consumers are also likely to consider their social status and maintain their image in society. This might persuade followers' reactions towards values promoted in PSI [45]. Therefore, PSI might be a tool to mitigate the negative online impacts of perceiving value perceptions in PSI

with Instagram influencers. PSI does not guarantee that negative social media issues will be directly mitigated; it depends on social media influencers sending strong messages and maintaining their image to benefit from business and followers. Because followers admire the attractiveness of Instagram influencers, they are considered role models for constructive social change [6].

Lastly, value perception of luxury brands strongly impacts purchase intention after PSI with Instagram influencers delivers those value perceptions to followers. Personal value is the greatest factor motivating consumer purchase intention of luxury brands, compared to social and conspicuous value. However, PSI with Instagram influencers cannot directly persuade followers to purchase luxury brands without the intermediary of value perceptions. When Thai millennials encounter the value perceptions of influencers, they might respond in purchase intention. Thus, reaction to value perceptions motivates followers' purchasing intentions, based on messages of influencers. For example, influencers sharing their opinions through persuasive captions and images can instigate reactions in their followers, as exemplified by the case of Thairath online news (2020) [91]. Instagram user @urassayas posted a photo with Monstera and a caption about her being happy with the decorative plant (#KeepCoolandStaySafe) during the COVID-19 lockdown pandemic, garnering the attention of a large number of followers. This led to increased prices for Monstera products and a high market demand due to this influencer's message, showing that an influencer can mitigate the negative aspects of social media through the use of their voice. Furthermore, because Thai people have a collectivist culture concerned with the perception of their personality in society, their decisions are based on what is best for the group [62].

Our findings contribute significantly to the existing body of knowledge on reducing the negative impacts of social media and the concept of PSI of Thai millennials regarding luxury product consumption. By mitigating the negative effects of social media marketing, the findings can be adopted to improve society by following, at first the power of online influencers can be used to build strong PSI between followers and influencers by prioritizing social and physical attractiveness and value perceptions (e.g., increasing perceiving personal value is the main focus for reducing online issues). Second, online influencers impact their followers by expressing messages through photos, videos, opinions, captions, and sharing content, which produces social activity and consumer reactions, as described above. Lastly, online influencers need followers; therefore, they benefit from expressing their personalities to social media users by producing content to engage potential followers.

Additionally, this research has managerial implications for luxury fashion companies that use influencer marketing; it might be used to reinforce a deep understanding of luxury consumption based on value perception to purchase intention. Both the social and physical attractiveness of influencers play a significant role in managers' selection of influencers to positively establish relationships with followers and increase the likelihood of future purchases. The manager should focus on the social attractiveness of influencers, such as their lifestyle or personalities, and their similarity to target consumers rather than their physical attractiveness. Establishing a partnership with customers through online influencer marketing can also improve sales. This study clarifies the relationship between online marketing tactics, brand sponsorship/brand ambassadors, and the characteristics of the target audience, such as their values, beliefs, and desires [45], and this information can be used when managers select an influencer for social media advertising to promote corporate sustainability [7]. Our findings support the use of online Instagram marketing and fashion influencers as a channel for fashion brand managers to build relationships with consumers outside of traditional marketing strategies. Moreover, PSI with influencers delivers their opinions to followers regarding value perceptions (social, personal, and conspicuous value). Managers should understand the power of PSI which can increase social activity via value perceptions of goods or brands. This research shows personal value is the most important factor in the purchase an expensive product. Additionally, social value and conspicuous value also reflect Thai society's concern with social status and image.

Thus, managers can develop marketing strategies via understanding PSI with influencers to deliver value perceptions. Additionally, consumer intention cannot motivate purchase of luxury brands without considering value perceptions from PSI because Thai millennials consider social, personal, and conspicuous value before buying an expensive brand. Thus, Thai followers should be engaged on social media platforms to stimulate perception of goods and determination of their value. Management can adapt this knowledge to develop marketing strategies that increase promotion of value perceptions by media personalities.

*Limitations and Future Research*

The generalizability of this study is limited to sustainable marketing, for a number of reasons. Firstly, this study only surveyed consumers and Instagram influencer interviews, which may lead to a potential attitude towards the value perception–behavior gap. Secondly, it was not clear when consumer intention to buy (via online platform or physical store) resulted from PSI on Instagram motivating their purchase intention. Third, the research studied the overall image of influencers, rather than differentiating levels of influencer; therefore, it should identify the main target of studying influencers for a greater impact. The sampling procedure of 400 Thai millennial respondents with convenience sample method; other methods such as snowball, judgmental, and probability sampling can be used in the future. Lastly, the present research only studies the relative positive impact of PSI, value perception, and purchase intention in the luxury fashion field; however some possible disadvantages may exist, such as the potential cultural issues arising from the use of counterfeit products of a luxury brand. These limitations invite future research opportunities. Future research may examine other possible determinants of the electronic word of mouth (eWOM) of the PSI in the context of luxury fashion. Therefore, future studies may also investigate consumers' perceptions of eWOM opinions on online platforms, such as Facebook and Instagram, and how they motivate the intention to buy luxury fashion. To build a strong relationship between the consumer and eWOM, studies might examine cultural differences in consumption of luxury fashion. In this way, better insights can be gleaned regarding appropriate marketing and promotion tactics. Additionally, further investigation can identify the drivers which lead consumers to express other behavioral outcomes, such as engagement on social media (seeking information and sharing opinions) and can employ different methodological approaches to design future studies to investigate the effects of the specific variables on luxury fashion behavior.

**Author Contributions:** Conceptualization, A.J. and S.P.; methodology, A.J.; software, A.J.; formal analysis, A.J. and S.P.; writing—original draft preparation, A.J.; writing—review and editing, S.P.; project administration, A.J.; funding acquisition, A.J. All authors have read and agreed to the published version of the manuscript.

**Funding:** This research was funded by Walailak University, grant number WU64201.

**Institutional Review Board Statement:** The study was approved by the Ethics Committee of Walailak University (WUEC-21-036-01) and date of approval 5 March 2021.

**Informed Consent Statement:** Informed consent was obtained from all subjects involved in the study.

**Data Availability Statement:** Data is contained within the article.

**Acknowledgments:** The authors express their gratitude for supporting this study to Walailak University and the Center of Excellence for Tourism Business Management and Creative Economy.

**Conflicts of Interest:** The authors declare no conflict of interest.

# Appendix A

**Table A1.** Analysis statistics of confirmatory factor analysis (CFA).

| Variable | Standardized Factor Loading | Error Variances | t-Value | R2 | AVE | CR |
|---|---|---|---|---|---|---|
| Social Attractiveness | | | | | 0.520 | 0.811 |
| SA1 | 0.670 | - | - | 45.0% | | |
| SA2 | 0.690 | 0.060 | 16.539 * | 48.0% | | |
| SA3 | 0.840 | 0.070 | 14.732 * | 70.0% | | |
| SA4 | 0.660 | 0.070 | 12.866 * | 44.0% | | |
| Physical Attractiveness | | | | | 0.560 | 0.835 |
| PA1 | 0.760 | - | - | 58.0% | | |
| PA2 | 0.680 | 0.060 | 12.066 * | 67.0% | | |
| PA3 | 0.820 | 0.090 | 10.581 * | 47.0% | | |
| PA4 | 0.720 | 0.080 | 9.728 * | 52.0% | | |
| Parasocial Interaction | | | | | 0.508 | 0.788 |
| PS1 | 0.650 | - | - | 42.0% | | |
| PS2 | 0.700 | 0.060 | 16.934 * | 49.0% | | |
| PS3 | 0.810 | 0.070 | 15.033 * | 65.0% | | |
| PS4 | 0.610 | 0.070 | 11.716 * | 38.0% | | |
| Social value | | | | | 0.578 | 0.846 |
| SV1 | 0.770 | - | - | 59.0% | | |
| SV2 | 0.720 | 0.060 | 15.310 * | 52.0% | | |
| SV3 | 0.780 | 0.020 | 45.593 * | 61.0% | | |
| SV4 | 0.770 | 0.060 | 17.278 * | 59.0% | | |
| Personal value | | | | | 0.518 | 0.810 |
| PV1 | 0.630 | - | - | 40.0% | | |
| PV2 | 0.730 | 0.090 | 11.929 * | 54.0% | | |
| PV3 | 0.720 | 0.080 | 12.085 * | 51.0% | | |
| PV4 | 0.790 | 0.100 | 12.573 * | 64.0% | | |
| Conspicuous value | | | | | 0.605 | 0.856 |
| CA1 | 0.840 | - | - | 71.0% | | |
| CA2 | 0.650 | 0.060 | 13.715 * | 42.0% | | |
| CA3 | 0.950 | 0.050 | 21.693 * | 90.0% | | |
| CA4 | 0.630 | 0.050 | 14.043 * | 40.0% | | |
| Purchase Intention | | | | | 0.627 | 0.870 |
| PI1 | 0.760 | - | - | 58.0% | | |
| PI2 | 0.720 | 0.070 | 14.691 * | 52.0% | | |
| PI3 | 0.870 | 0.060 | 20.102 * | 76.0% | | |
| PI4 | 0.800 | 0.070 | 15.638 * | 64.0% | | |

* Sig. < 0.001.

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
