# Peer review of "How Instagram Influencers Affect the Value Perception of Thai Millennial Followers and Purchasing Intention of Luxury Fashion for Sustainable Marketing"

_sustainability, doi:10.3390/su13158572_

Round 1
Reviewer 1 Report
Manuscript ID: sustainability-1279171, entitled "How Instagram InfluencersAffect Thailand 2 Millennial Lovers' Value Perception and Luxury Fash-3
Emerging Market Marketing Purchasing Intent", has been reviewed. . This is a well-executed study. For me, there are no key issues in
the manuscript. The author only needs to consider four points: Just as McDonald and Ho (2002) suggested that authors better
provide a "sample co-matrix" in the manuscript or appendix for
readers or other researchers to reanalyze. The numbers in the
matrix should be kept to the third decimal place.
Because the author(S) is too, but "** sig". <Format table 4,
why why sig. In H1, 0.009<0.05 only one*, in H2...4c? The reviewer suggested that the author can push * or ** to Figure 2. This is the discussion in Section 4. Although the author included the
results in Section 4, there are tables and figures, none of which
indicate whether the hypothesis is valid. The author also concludes
the overall discussion in Section 5 without explaining whether
the false is true.
And readers, like reviewers, can't get people's
discovery and truth the first time. Therefore, the reviewer
recommends reading a simple summary of each hypothetical
study and explaining item by item before the conclusion
for the convenience of readers. Would be more appropriate.
Author Response
Dear Editor in Chief, Guest Editors, and Anonymous Reviewers,
Thank you very much for allowing us to revise our paper and providing valuable suggestions. We have carefully gone through the feedback and have revised our paper accordingly. These suggestions have certainly helped us to significantly improve the quality of our submission. We have provided pointwise responses to all queries below and have used track changes function in the paper to show the modifications.
We sincerely hope that our revised paper will meet your expectations. We look forward to hearing the views on our paper revised paper soon.
Kind Regards
All co-authors

Reviewer 2 Report
This paper deals with an interesting topic for both scholars and practitioners: The impacts of influencers on consumers PSI (parasocial interaction) for luxury products.
However, I suggest a major revision on the following grounds.
1) I do not see how this paper answers its main objective, that is, demonstrate how PSI mitigates the negative impacts of social networks. My understanding is that PSI, is a mediator between social variables and value perception. Nothing highlights a 'mitigating effect'.
2) Hypotheses need to be better introduced.
3) The sampling procedure should be described. Is it a convenience sample? Then its limitations should be discussed.
4) The scales are not introduced. Were they adapted from existing scales? Besides, did the author do a back translation?
5) Results should be presented properly. Paragraph 4.4. should present the results for each hypothesis and I suggest Table 4 also clearly indicates which hypotheses are validated and which hypotheses are not validated.
6) The discussion needs to be re-written and structured consistently with the hypotheses testing. it should also enhance the theoretical implications of this research, as well as the implications for managers. The data suggest that H% is not validated. How can we explain this very counter-intuitive result?
7) Importantly, the paper does not enhance sufficiently the importance of studying luxury products here.
We believe this paper could be considerably improved and wish the authors the best of luck with this paper.
Author Response
Dear Editor in Chief, Guest Editors, and Anonymous Reviewers,
Thank you very much for allowing us to revise our paper and providing valuable suggestions. We have carefully gone through the feedback and have revised our paper accordingly. These suggestions have certainly helped us to significantly improve the quality of our submission. We have provided pointwise response to all queries below and have used track changes function in the paper to show the modifications.
We sincerely hope that our revised paper will meet your expectations. We look forward to hearing the views on our paper revised paper soon.
Kind Regards
All co-authors

Reviewer 3 Report
This paper intends to establish and extending using PSI as an moderator to mitigate the negative effect of social media by examining the relationship between PSI and followers of luxury fashion’s value (social, personal, and conspicuous) on Instagram.
It is well written, literature was thorough and the research method was explained in detail. The discussion and conclusion were supported from the data and analysis. However, I am confused with the contribution of this research to the relevancy of this journal. I am not very convinced with the sustainability perspective of this study. The authors may either rewritten the contribution part to make it closely align with the objectives of this journal or submit to Journals related to consumer behavior or social marketing.
Author Response

(The authors gave the same response as above.)

Round 2
Reviewer 2 Report
Dear author(s)
This paper has improved on some points (sample description; presentation of the results and discussion).
However, I suggest a major revision on the following grounds.
1) I do not see how this paper answers its main objective, that is, demonstrate how PSI mitigates the negative impacts of social networks. My understanding is that PSI, is a mediator between social variables and value perception. Nothing highlights a 'mitigating effect'.
2) Hypotheses need to be better introduced. They are presented altogether (but H5 which is well introduced). Each hypothesis needs to be introduced (i.e. suggested based on previous research and theoretical grounds) separately.
3) The scales are not introduced. Which authors or previous scales do you rely on? You should precise it in the corpus text of the methodology. Besides, did the author do a back translation? do not see any answer to this question.
4) The discussion needs to be proofread, and maybe more clearly organised while presented the main finding and then less important ones.
5) Importantly, the paper does not enhance sufficiently the importance of studying luxury products here, in the discussion part.
We believe this paper should be improved to be published and wish the author(s) the best of luck with this paper.
Author Response
Dear Editor in Chief, Guest Editors, and Anonymous Reviewers,
Thank you very much for allowing us to revise our paper and providing valuable suggestions. We have carefully gone through the feedback and have revised our paper accordingly. These suggestions have certainly helped us to significantly improve the quality of our submission. We have provided pointwise responses to all queries below and have used the track changes function in the paper to show the modifications.
We sincerely hope that our revised paper will meet your expectations. We look forward to hearing the views on our paper revised paper soon.
Kind Regards
All co-authors
